# A Clinical Case of Nosocomial Pneumonia as a Complication of COVID-19: How to Balance Benefits and Risks of Immunosuppressive Therapy?

**DOI:** 10.3390/antibiotics12010053

**Published:** 2022-12-29

**Authors:** Svetlana Rachina, Gairat Kiyakbaev, Elena Antonova, Alexey Mescheryakov, Olga Kupryushina, Girindu Hewathanthirige, Ivan Palagin, Elena Kozhevnikova, Marina Sukhorukova, Daria Strelkova

**Affiliations:** 1Internal Medicine Department #2, Sechenov First Moscow State Medical University, 119991 Moscow, Russia; 2War Veterans Hospital #3 of Moscow, 129336 Moscow, Russia; 3Medical Faculty, Peoples’ Friendship University of Russia, 117198 Moscow, Russia; 4Institute of Antimicrobial Chemotherapy, Smolensk State Medical University, 214019 Smolensk, Russia; 5N.N. Burdenko National Medical Research Center for Neurosurgery, 121087 Moscow, Russia

**Keywords:** COVID-19, immunosuppressive therapy, nosocomial pneumonia, dexamethasone, carbapenemase

## Abstract

We report a Russian case of a 61-year-old male patient with confirmed COVID-19 infection who developed nosocomial pneumonia complicated by lung abscess associated with multi-drug-resistant isolates of *Klebsiella pneumoniae* and *Acinetobacter baumannii*, which could have been provoked due to the immunosuppressive therapy. We discuss the existing literature highlighting the issue of the prudent balance between benefits and risks when prescribing immunomodulators to hospitalized patients with COVID-19 due to the risk of difficult-to-treat nosocomial infections caused by MDR Gram-negative bacterial pathogens. Currently, there is evidence of a substantial positive effect of dexamethasone on the course of COVID-19 in patients requiring supplemental oxygen or anti-interleukin-6 drugs in individuals with prominent systemic inflammation. However, it seems that in real clinical practice, the proposed criteria for initiating treatment with immunomodulators are interpreted arbitrarily, and the doses of dexamethasone can significantly exceed those recommended.

## 1. Introduction

Even before the COVID-19 pandemic, an increase in the prevalence of multi-drug resistant (MDR) Gram-negative bacteria was considered a global health threat [1]. According to available reports, carbapenem-resistant Enterobacterales (CRE), especially *Klebsiella pneumoniae* and MDR *Acinetobacter baumannii*, are increasingly being detected as nosocomial pathogens in Russian hospitals [2,3,4].

During the current pandemic, a trend toward expansion of MDR Enterobacterales and nonfermenting Gram-negative bacilli, as well as a change in the epidemiology of carbapenemases produced by *K. pneumoniae*, was noted [4,5,6,7]. We assume that excessive or inappropriate prescription of antibiotics, use of glucocorticosteroids (GCS) and immunomodulatory therapy, along with weakening of infection control, can be the main triggers of it.

During the first year of the COVID-19 pandemic, 69% of patients reported using antibiotics prior to hospital admission [8]. In Russian COVID-19 hospitals, an average prevalence of antibiotic use was 35.1% and varied from 29.7% in medical wards up to 75.6% in intensive care units [9]. Patel A. et al. showed that high antibiotic use, along with critical illness, double occupancy of a single room, and modified infection prevention practices, were the key factors contributing to the rapid spread of MDR Gram-negative bacteria in COVID-19 care units of a hospital in Maryland, USA [10]. In a propensity-matched cohort study, Scaravilli V. et al. demonstrated that critically ill COVID-19 patients are at high risk for ventilator-associated pneumonia caused by MDR bacterial pathogens, which is attributed to the use of GCS [11].

Here, we present a clinical case of a patient with confirmed COVID-19 infection who developed nosocomial pneumonia (NP) complicated by lung abscess associated with MDR isolates of *K. pneumoniae* and *A. baumannii*, leading to a significant prolongation of his hospital stay.

## 2. Case Presentation

A 61-year-old male, BMI of 30.9 kg/m^2^, with no history of smoking, drinking, or significant comorbidities, was admitted to the therapeutic department with a diagnosis of moderate COVID-19 (PCR-detected SARS-CoV-2 virus), complicated by bilateral pneumonia on chest-computed tomography (CT), with involvement of both lungs up to 25%. The main symptoms developed 5 days before the hospitalization and were general weakness, malaise, and dry cough. Before the admission, he took amoxicillin/clavulanate, levofloxacin, favipiravir, and rivaroxaban prescribed by a GP.

His general condition upon admission was of moderate severity: body (axillar) temperature of 38 °C, respiratory rate (RR) of 22/min, oxygen saturation (SpO_2_) of 94%, heart rate at 85 beats/min, and blood pressure of 130/70 mm Hg. His laboratory tests revealed mild thrombocytopenia and an increase in serum lactate dehydrogenase (LDG) (Figure 1). Favipiravir was continued, ambroxol, nadroparin, and dexamethasone were added, and low-flow oxygen at flow rate of 5 L/min was initiated.

On day 3, the patient’s condition worsened; he experienced progressive dyspnea and tachypnea (RR 26/min), requiring a switch to high-flow oxygen therapy. The second CT scan revealed a significant increase in involvement of parenchyma of both lungs up to 75% (Figure 2a). The dexamethasone dose was increased from 8 to 24 mg/day.

On day 5 of the hospital stay, due to the increase in LDG levels and lymphopenia (0.8 × 10^9^/L) regarded as a “cytokine storm” manifestation, the patient was administered a single dose of levilimab, 324 mg, intravenously.

Since day 7, the patient’s condition began to progressively worsen due to respiratory failure (RF); he was transferred to the intensive care unit (ICU) and switched to non-invasive mechanical ventilation in PEEP/CPAP mode. The chest CT showed an increase of bilateral parenchymal involvement area up to the critical severity (Figure 2b). Sputum and blood cultures were taken and did not reveal any clinically significant pathogens.

On day 12, leukocytosis with neutrophil left shift and elevation of serum inflammatory markers, along with the appearance of purulent sputum and fever relapse (38.7 ’C), were recorded, suggesting the development of NP, so meropenem and linezolid were empirically prescribed. The patient’s condition was complicated with the development of anemia and hypoalbuminemia and remained severe (Figure 1).

On day 17, a sputum culture revealed growth of *K. pneumoniae* (10^8^ CFU/mL), resistant to ampicillin, amoxicillin/clavulanate, amikacin, meropenem, ertapenem, cefotaxime, ceftazidime, ciprofloxacin, and co-trimoxazole. Linezolid was discontinued, polymyxin B was added to the treatment, and the dose of meropenem was increased up to 6 g/day by prolonged infusion. Due to the lack of a significant effect after 72 h, meropenem was discontinued, and tigecycline at a dose of 200 mg/day was added to polymyxin B.

On day 35, despite a slight improvement of laboratory inflammatory markers, the patient developed acute kidney injury (AKI), with the glomerular filtration rate of 20 mL/min/1.73 m^2^. Intermittent renal replacement therapy (RRT) was initiated. The chest CT showed a previously non-visualized cavity with a fluid level in the right lung, which may have corresponded to a lung abscess (Figure 2c). A repeated sputum culture revealed the same phenotype of MDR isolate of *K. pneumoniae* (10^5^ CFU/mL), producing OXA- and NDM-like carbapenemases, which was confirmed by immunochromatographic NG-Test CARBA5 (NG Biotech, France). Polymyxin B and tigecycline were discontinued, with simultaneous prescription of ceftazidime/avibactam and aztreonam.

On day 44, positive clinical and laboratory dynamics were noted. However, the patient had persistent RF and purulent sputum. *A. baumannii* (10^5^ CFU/mL), isolated from the sputum, was resistant to ampicillin/sulbactam, amikacin, meropenem, ceftazidime, ciprofloxacin, and co-trimoxazole. Thus, tigecycline was resumed at a dosage of 200 mg/day.

By day 58 of the hospital stay, the patient’s condition improved significantly, and normalization of laboratory inflammatory markers was noted; the chest CT revealed a decrease in infiltration and cavity size (Figure 2d), and the sputum culture revealed no growth. Antibiotic therapy was discontinued. The patient was transferred to the therapeutic unit, and on day 69, he was discharged from the hospital.

A follow-up chest CT 2.5 months after patient’s discharge showed residual post-inflammatory changes in both lungs (Figure 2e). The patient returned to a normal lifestyle and reported no complaints.

## 3. Discussion

Although bacterial co-infections are generally uncommon, the frequency of superinfections in COVID-19 represents a major problem among hospitalized patients. Available reports range the rate of bacterial nosocomial infections (NI) from 5 to 21%, and they can even exceed 40% in the ICU [12,13,14,15]. In the structure of NI in hospitalized patients with COVID-19, the bloodstream infections and pneumonia prevail [14,15]. Llitjos J.-F. et al. showed COVID-19 is an independent risk factor for the development of ICU-acquired pneumonia [16].

The spectrum and antimicrobial susceptibility of bacterial pathogens causing NI in patients with COVID-19 generally reflect the profile of pathogens circulating in a particular hospital/department [4,17]. Bacterial superinfections are associated with a significant increase in mortality and the length of hospital stay [18,19].

The diagnosis of NP in severe COVID-19 is difficult, as new infiltrates are poorly visualized in patients with widespread lung damage; leukocytosis can be a demonstration of the known pharmacodynamic effect of systemic GCS, and the use of dexamethasone and interleukine-6 (IL-6) antagonists is associated with a significant decrease in C-reactive protein and procalcitonin and/or the absence of their increase in response to bacterial superinfection [20,21].

This clinical case raises a number of concerns related to the diagnosis and treatment of patients with COVID-19, specifically the need to hospitalize patients with non-severe infection, taking into account the risk of exposure to nosocomial MDR pathogens. Apparently, a careful balance of benefits and risks is required when dexamethasone or other immunomodulators are prescribed. SARS-CoV-2 can activate an innate and adaptive immune system, resulting in the massive inflammatory response. This uncontrolled inflammation may lead to systemic tissue damage and coagulopathy [22,23].

Currently, there is evidence of a substantial positive effect of dexamethasone on the course of COVID-19 in patients requiring supplemental oxygen or anti-IL-6 drugs in individuals with prominent systemic inflammation [24,25]. 

However, it seems that in real clinical practice, the proposed criteria for initiating treatment with immunomodulators are interpreted arbitrarily, and the doses of dexamethasone can significantly exceed those recommended.

Our patient was prescribed dexamethasone from the moment of hospitalization despite the lack of obvious indications (SpO_2_ was 94%) at a starting dose of 8 mg, followed by dramatic increase to 24 mg. On day 5, levilimab—a recombinant monoclonal antibody to the IL-6 receptor—was administered due to the progression of RF but with no clear evidence of systemic inflammation (Figure 1). The worsening of the patient’s condition within the first week of the hospital stay was highly likely to have been associated with the progression of COVID-19 and critical lung damage. However, such “aggressive” immunosuppressive therapy cannot rule out its contribution for NP and lung abscess development.

It should be emphasized that both GCS and anti-IL-6 drugs increase the likelihood of various adverse reactions [26,27]. In an updated systematic review and meta-analysis, the same authors revealed an increased risk of neutropenia, impaired liver function, and secondary infections among patients with COVID-19 receiving IL-6 receptor antagonists, showing that the beneficial effects of IL-6 receptor antagonists were observed in patients with a high baseline risk of mortality, emphasizing the need for a balanced approach to their prescription [28]. In a multicentre observational study conducted in Norway, dexamethasone use was strongly and independently associated with superinfections among critically ill COVID-19 patients [29]. It should be noted that the specific mechanisms contributing to superinfections when prescribing GCS or IL-6 receptor antagonists to patients with COVID-19 are not completely clear. However, similar pharmacodynamic effects of these drugs were shown even before the COVID pandemic and are unlikely to differ in this specific group of patients.

Another important issue is the need for early initiation of adequate antimicrobial therapy of bacterial superinfections. In our case, in accordance with the local hospital guidelines, the patient was empirically prescribed meropenem with linezolid, followed by the addition of polymyxin B and the replacement of meropenem by a high dose of tigecycline. Unfortunately, minimal inhibitory concentration of meropenem was not measured, and the type of carbapenemases produced by *K. pneumoniae* was determined by day 35 only, giving evidence for prescription of ceftazidime/avibactam and aztreonam combination with the subsequent addition of tigecycline, since MDR *A. baumannii* superinfection was detected.

It is important to emphasize the majority of therapeutic options for severe CRE infections in the ESCMID guidelines (e.g., meropenem/vaborbactam and cefiderocol) are not available in Russia, so combined antibiotic therapy with ceftazidime/avibactam and aztreonam in case of NP caused by co-producers of OXA-like and NDM carbapenemases is considered the first line in the national guidelines [30]. It should be noted that during treatment with polymyxin B, the patient developed AKI, requiring its discontinuation and initiation of RRT.

Given the limited data of outcomes in COVID-19 patients with critical lesions of the lung parenchyma, complicated by the development of NP and lung abscess, the dynamics of chest CT findings deserve special attention, especially the complete regression of the lung changes in 2.5 months.

## 4. Conclusions

In our clinical observation, we present a comparatively common case of superinfection in a hospitalized patient with COVID-19. However, it highlights the need to balance benefits and risks of immunosuppressive therapy: on the one hand, a careful patient selection and timing for dexamethasone and IL-6 receptor antagonists provides potential benefits in case of the development of prominent hyperimmune response, but on the other hand, it increases the risk of side effects, including difficult-to-treat NP caused by MDR gram-negative bacterial pathogens.

## Figures and Tables

**Figure 1 antibiotics-12-00053-f001:**
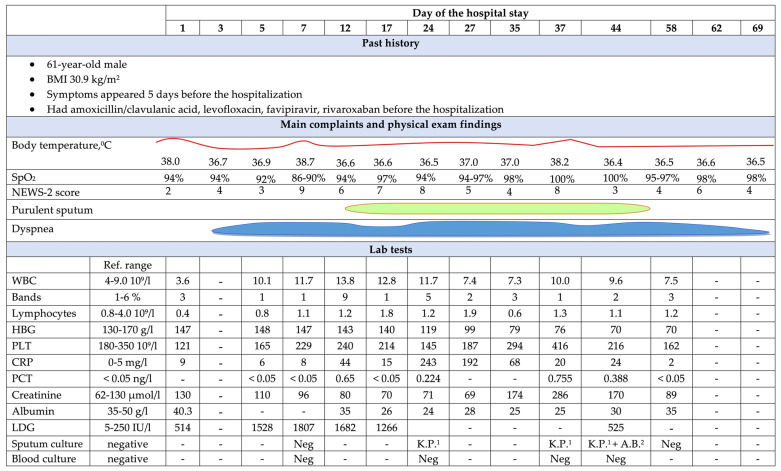
Clinical course of the disease in a 61-year-old male patient. ^1^ K.P.—*K. pneumoniae*; ^2^ A.B.—*A. baumannii*; ^3^ loading dose was 200 mg; WBC—white blood cells, HBG—hemoglobin, CRP—C-reactive protein, PCT—procalcitonin, PLT—platelets, LDG—lactate dehydrogenase, CT—computer tomography, TID—three times per day, BID—two times per day, PO—orally, IV—intravenously, aPTT—activated partial thromboplastin time, LFO—low flow oxygen, HFO—high flow oxygen, NIV—non-invasive ventilation, SC—subcutaneously, IU—international units.

**Figure 2 antibiotics-12-00053-f002:**
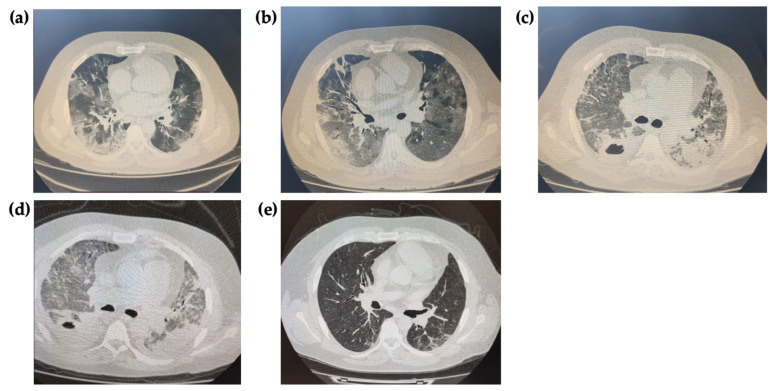
(**a**) Day 3. Bilateral polysegmental infiltration (ground glass opacity), a high probability of viral origin of pneumonia with involvement of lung parenchyma from 50 to 75%; (**b**) Day 7. An increase in involvement of parenchyma of both lungs up to 75% due to ground glass type interstitial infiltration, the appearance of perivascular and subpleural consolidation lesions in the right lung; (**c**) Day 35. An increase of the parenchyma infiltration of both lungs due to ground glass opacity up to 95%, previously non-visualized limited air cavity in S_6_ of the right lung with thick uneven walls, with a fluid level in the lumen (45 × 30 × 36 mm), most likely corresponding to lung abscess; (**d**) Day 58. The trend towards a decrease in size of destructive cavity; (**e**) Follow up chest CT 2.5 months after the hospital discharge. Residual post-inflammatory changes in the form of reticular changes, single bullae up to 8 mm in diameter.

## Data Availability

The data used during the current study are available from the corresponding author (S.R.) on request.

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
