# Peer review of "A Clinical Case of Nosocomial Pneumonia as a Complication of COVID-19: How to Balance Benefits and Risks of Immunosuppressive Therapy?"

_antibiotics, 2022, doi:10.3390/antibiotics12010053_

Round 1

Reviewer 1 Report

I feel that this case, even if it is well written, is of very little use to readers. The picture of pneumonia could be totally attributable to COVID and the isolated microorganisms could only be colonizers

Reviewer 2 Report

The case study presented by the authors is very interesting as it addresses the need for attention to nosocomial infections in COVID-19 especially as it is now endemic and would be presented as a good study for future treatment. I have a few suggestions that could be considered by authors.

1. Case presentation could be separated to "case presentation" and "management and outcome" in order to help readers to better understand.

2. There are some formatting error e.g. line 151

3. Cytokine storm is normally the cause of damaged tissues in COVID-19. Can the authors discuss about it in the discussion as it was only mentioned in case presentation but not in the discussion.

Thank you

Reviewer 3 Report

Authors have presented only 1 case of nosocomial pneumonia (NP) in COVID-19 and hence it is possible authors observations could be improved with examining more cases and comparing with cases where NP was not seen. If authors cannot include more studies or literature comparison, the manuscript title should reflect the speculation in this study.

Lines 38-41 = Authors assume effect of excessive or inappropriate prescription of antibiotics including others are the main trigger of NP. A literature search might give more examples which support or counter author insights. 

Lines 144-145 = Could authors provide possible mechanism of "aggressive" immunotherapy induced contribution for NP and lung abscess development.

Round 2

Reviewer 1 Report

The paper has been improved and can now be accepted

Reviewer 3 Report

Authors have satisfactorily answered my comments.